# Viral load suppression and its predictor among HIV seropositive people who receive enhanced adherence counseling at public health institutions in Bahir Dar, Northwest Ethiopia. Retrospective follow-up study

**Minyichil Birhanu Belete**[1]*, **Abebayehu Bitew**[2☉], **Kebadnew Mulatu**[2☉]

**1** Department of Pediatrics and Child Health Nursing, School of Health Sciences, College of Medicine and Health Sciences, Bahir Dar University, Bahir Dar, Ethiopia, **2** Department of Epidemiology and Biostatistics, School of Public Health, College of Medicine and Health Sciences, Bahir Dar University, Bahir Dar, Ethiopia

☉ These authors contributed equally to this work.
* mbym24@gmail.com, minyichil.birhanu@bdu.edu.et

**Data Availability Statement:** All relevant data are within the manuscript and its Supporting Information files.

## Abstract

### Background

For those HIV seropositive people with high viral loads, the World Health Organization recommends more counseling before changing ART regimens. A high viral load can lead to increased HIV transmission and lower survival rates. Clients with viral loads above 1000 copies/mL should receive enhanced adherence counseling for 3–6 months before switching. Despite enhanced adherence counseling programs, most countries struggle with viral load suppression. Little is known about viral load suppression in Ethiopia and the research area after counseling.

### Objective

This study aims to assess viral load suppression and its predictors among HIV-positive individuals receiving enhanced adherence counseling in Bahir Dar, Northwest Ethiopia, in 2022.

### Methods

An institution-based retrospective follow-up study was conducted among randomly selected 546 clients on Enhanced Adherence Counseling at public health facilities in Bahir Dar city. The Epicollect5 mobile application was used to collect the data, which was then exported to Stata version 14 for analysis. A Log-Binomial regression model was fitted for each explanatory variable. Variables having a p-value <0.25 in bivariate analysis were entered into a multivariable Log-Binomial regression model. Finally, an adjusted risk ratio with a 95% confidence interval and a p-value <0.05 was used to measure the strength of the prediction.

**Funding:** For this research, the author(s) received no specific funding.

**Competing interests:** The authors declare that they have no competing interests.

**Abbreviations:** ARR, Adjusted Risk Ratio; ART, Anti-retroviral Therapy; EAC, Enhanced Adherence Counseling; HIV, Human Immune Deficiency Virus; OI, : Opportunistic Infection; PLHIV, People Living with HIV; UNAIDS, United Nations Program for HIV/AIDS; WHO, World Health Organization.

## Results

Following enhanced adherence counseling, 312 (57.1%) people had their viral load suppressed. Absence of recurrent OI (ARR 1.40; CI 1.03–1.91), EAC stay less than 3 months (ARR 1.54; CI 1.19–1.99), EAC stay 3–6 months (ARR 1.38; CI 1.12–1.69), once-daily ARV dose regimen (ARR 1.28; CI 1.03–1.58), baseline viral load of 2879.00 copies/ml (ARR 1.30, CI 1.06–1.60), being orthodox Tewahido Christian (ARR 0.37; CI 0.18–0.75) were significant predictors of viral load suppression after Enhanced Adherence Counseling.

## Conclusion and recommendation

Most importantly, this study found that most people had suppressed viral loads after receiving enhanced adherence counseling. Significant predictors of viral load suppression included recurrent OI, length of stay on EAC, daily ARV dosing regimen, baseline viral load, and religion. Clients with a high baseline viral load and those who experience recurring opportunistic infections should get extra care during EAC sessions.

## Introduction

Globally, since the beginning of the human immunodeficiency virus (HIV) epidemic in the 1980s, more than 77.5 million people have acquired the infection, and about 34.7 million people have died [1]. In 2020, 37.6 million people were living with HIV, 1.5 million were newly infected and 690,000 people died from acquired immunodeficiency syndrome (AIDS) related illnesses [2]. However, the burden of HIV epidemics varies considerably throughout the world, with 67% of the global burden concentrated in the African region [3].

In people living with the human immune deficiency virus (PLHIV), viral load (expressed as HIV RNA copies/mL of blood) is a direct indicator of viral replication. Higher viral loads lead to a greater fall in CD4 cell count, and this increases the risk of becoming ill due to opportunistic infections [4]. Suppressing the viral load in PLHIV to less than 1000 copies/ml of blood (henceforth called "viral suppression") is essential for reducing morbidity, mortality, and transmission [5].

The goal of antiretroviral therapy for HIV infection is to achieve and maintain virologic suppression, thereby preventing disease progression and transmission. The World Health Organization (WHO) recommended viral load testing as the preferred monitoring approach to diagnose and confirm treatment failure. This is because a decrease in CD4 count, which is a marker of immunological failure, occurs as a result of viral replication, which can thus be considered an endpoint [6]. Anti-retroviral therapy (ART) suppresses HIV replication, and by doing so, it has transformed HIV infection from a deadly disease into a manageable chronic illness [4, 7]. A recent clinical trial has shown that viral suppression due to ART can reduce HIV transmission by up to 96% [8]. To exploit the benefits of ART globally, the second and third targets of the joint United Nations Program for HIV/AIDS (UNAIDS) 95-95-95 ambitious plan target call for at least 95% of PLHIV to be on ART and 95% of those on ART should achieve a suppressed viral load by 2025 [9].

Viral load testing, in conjunction with a treatment monitoring algorithm is used to detect treatment failure, provide timely adherence interventions, and identifying potential drug resistance, promoting regimen adjustments if necessary [10, 11]. WHO recommends routine viral load testing at 6 and 12 months post-ART initiation, then annually thereafter to proactively identify treatment failure [4, 11, 12]. Enhanced Adherence Counseling (EAC) is recommended

for individuals with high viral loads (> 1000copies/mL), involving personalized plans to address adherence barriers and improve treatment outcomes [13, 14]. Clients with persistent high viral loads despite enhanced adherence counseling may require switching to a second-line or third-line regimen as per WHO guidelines, particularly if two consecutive viral load measurements within 3–6 months remain elevated [4, 10, 12]. Individuals with mental health problems or substance abuse issues might require extended support before regimen adjustments for treatment failure [4, 7].

Even though UNAIDS has set the three 95s (95-95-95) targets for 2025 [15], the WHO reported a gap in achieving its 95-95-95 global targets. In 2022, these percentages were 86%, 89%, and 93%, respectively [9]. Whereas, among all people living with HIV, 86% knew their status, 76% were accessing treatment, and 71% were virally suppressed in 2022 [16]. At the end of 2016, at least 19 million people living with HIV globally had started ART [17]. Of these, 72% were living in sub-Saharan Africa [18]. The public health impact of this achievement will depend on the extent to which those initiating ART are able to achieve and maintain virologic suppression [19].

The World Health Organization, based on its systematic review result recommended adherence interventions for clients with high viral loads [4]. Even though EAC has been implemented since 2016, suppression rates were low among ART-treated children with virological failure that completed the recommended three EAC sessions [20]. More than half (53.9%) of patients with an initial elevated viral load remained unsuppressed after enhanced adherence counseling, with only 53.4% appropriately switched to a new regimen. Suppression rates were higher among people with enhanced adherence counseling, but lower among children and adolescents compared to adults [21].

Worldwide, different studies reported that the magnitude of the unsuppressed viral load after enhanced adherence counseling ranged from 36% to 77% [20, 22–28]. The study in Ethiopia showed that the overall viral load suppression after enhanced adherence counseling ranged from 46.1 to 66.4% [14, 21]. Different studies showed that different factors contributed to viral load suppression, including sociodemographic characteristics, behavioral characteristics, clinical-related factors, and treatment-related factors [14, 21–26, 28–47].

Moreover, to our knowledge, scanty research has been done in Ethiopia, and no research has been found on a study area regarding viral load suppression after EAC sessions and its predictors among high viral load HIV seropositive individuals on ART. Existing studies suggest that achieving viral load suppression post-EAC remains below WHO targets, with contradictory findings on key predictor variables. This study aims to evaluate viral load suppression and predictors among HIV seropositive individuals receiving EAC at public health institutions in Bahir Dar City from January 2017 to December 2021, providing valuable insights for the Ethiopian Ministry of Health and other stakeholders to enhance the implementation of national guidelines and improve patient quality of life.

## Materials and methods

### Study design, setting, and period

An institution-based retrospective follow-up study was conducted among HIV seropositive people who receive enhanced adherence counseling at public health facilities in Bahir Dar city from January 2017 to December 2021, and the actual data extraction period was from January 1 to 30, 2022. Bahir Dar city has three public hospitals (Tibebe-Ghion Specialized Teaching Hospital, Felege-Hiwot Comprehensive Specialized Hospital, and Addisalem Primary Hospital) and six health centers (Bahir Dar Health Center, Shimbit Health Center, Dagmawi Minilik Health Center, Han Health Center, Shumabo Health Center, and Abay Health Center). All these public health facilities provide ART and enhanced adherence counseling services.

## Target population

The target population was all HIV seropositive people received enhanced adherence counseling at public health facilities in Bahir Dar city from January 2017 to December 2021.

## Study population

All selected HIV seropositive people who received enhanced adherence counseling at selected public health institution in Bahir Dar city

## Inclusion and exclusion criteria

All HIV seropositive people with documented high viral load and who received enhanced adherence counseling at Bahir Dar city public health institutions during the study period were included. Those HIV seropositive clients who had not finished enhanced adherence counseling sessions and clients who had no second viral load result were excluded from the study.

## Sample size and sampling procedures

The sample size was determined by using a single population proportion formula $(n = \frac{((Z_{\frac{\alpha}{2}})^2) \times p(1-p)}{d^2})$. Considering 95% Confidence Interval (CI) (Zα/2 = 1.96), 5% margin of error and the proportion of the client with estimated overall viral load suppression after enhanced adherence counseling among HIV seropositive was taken as 66.4% [14], generated a minimum sample size of 378 for the study. By using 1.5 design effect a total of 567 study populations were involved.

A simple random sampling using a computer generating system through SPSS version 26 software was used to select the study subjects in each health institution. Based on the number of clients who attend EAC sessions in each health institution, the proportional allocation of the total sample size was carried out to attain the required sample size in each health institution.

## Variables

**Dependent variable.** Viral load (suppressed/unsuppressed).

**Independent variables. Socio-demographic predictors:** Age, sex, residency, religion, educational status, marital status, type of health institution

**Baseline clinical and laboratory-related predictors:** Current ART regimen, baseline viral load count at the start of EAC session, WHO clinical stage, baseline CD4 count, nutritional status, opportunistic infections (OIs), functional, partner HIV status, condom utilization, and hospital-admission history

**Behavioral related predictors:** Cigarette smoking, alcohol drinking, khat chewing, disclosure status

**Treatment-related predictors:** Duration on ART, level of ART adherence, time gap from high viral load detected to EAC session start, time to complete EAC sessions, history of drug discontinuation

## Operational definition

**Enhanced Adherence Counseling:** A structured counseling intervention given monthly, for consecutive 3–6 times, to HIV seropositive people with a high viral load (>1000 copies/mL) with the aim of supporting them to achieve viral suppression

**High Viral Load**: Viral load of >1000 copies/ml on a routine or need-based viral load test.

**Baseline Viral Load**: Viral load count at start of EAC

**Suppressed Viral Load**: Viral load count ≤1000 copies/ml after EAC.

**Unsuppressed Viral load**: Viral load count >1000 copies/ml after EAC.

**Baseline CD4**: CD4 count at start of EAC

**Baseline Clinical stage**: WHO clinical stage at start of EAC

**Recurrent opportunistic infection**: Infections that occur more frequently or more severely in people with HIV than in people with healthy immune systems.

**Depressed:** If the client answer "Yes" for both of the following question;

Was there ever a time when you felt sad or hopelessness for more than 2 weeks in the last 3 month?

Was there ever a time lasting more than 2 weeks when you lost interest in most things like hobbies, work, or activities that usually give you pleasure in the last 3 month?

## ART adherence level

**Good ART adherence**: Average adherence ≥ 95% (Patient missing 1 from 30 doses or ≤3 doses out of 60 doses).

**Fair ART adherence**: Average adherence 85–94% (Patient missing 2–4 doses out of 30 doses or 4–9 doses from 60 doses).

**Poor ART adherence:** Average adherence < 85% (Patient missing ≥5 doses from 30 doses or ≥10 doses from 60 doses).

## Data collection procedures and quality control

Data were collected from the patient's chart, enhanced adherence counseling sheet, viral load registration book, and laboratory request using a pretested structured checklist. Accordingly, all charts containing detailed information about patients who were on ART were reviewed. If incomplete data is encountered, the data collectors was try to get the information from different data sources (patient's chart and follow-up form). Additionally, if clinical parameters and laboratory results (CD4 count and WHO clinical stage) were not found at the start of enhanced adherence counseling sessions, the data which were most recent to the starting date of enhanced adherence counseling sessions was considered as baseline data. Three Bachelor Science degree nurses as data collectors and three ART data managers as supervisors was participate in the data collection process.

Data quality was assured through designing a proper data collection tool. Training was given for both data collectors and supervisors on the objective of the study, data extraction, recording forms patient's charts, registration books using Epicollect5 mobile application for one day. Pretest was done by taking 5% of the overall sample at Felege-Hiwot Comprehensive specialized Hospital. After pretest tool modification was made by adding variables like spouse HIV status and removing some variables like income and smoking status. During the data collection period, a supervisor was assigned to make sure that there were no missed data. The overall activities of data collection was controlled by the principal investigator of the study.

## Data handling and analysis

Data were collected with Epicollect5 mobile android application and directly exported to Excel and then to STATA version 14 for analysis. Then data was cleaned by observing the frequency, cross-tabulation tables, and sorting to check missed values and outliers. Descriptive statistics, including mean, median, standard deviation, interquartile range, frequencies tables, and graphs was used to describe the characteristics of the study participants. Viral load suppression after EAC sessions was presented as proportion.

Bivariable Log-Binomial regression model was fitted for each explanatory variable. Multi-collinearity between the independent variables were checked using correlation matrix and variance inflation factor. Variables with a p-value <0.25 were a candidate variable for a multivariable Log-Binomial regression model to identify statistically significant predictors of viral load suppression by adjusting for possible confounders. Finally, Adjusted Risk Ratio (ARR) with a 95% confidence interval and p-value <0.05 was used to measure the strength of predictors. Hosmer-Lemeshow statistic tests was used to test for model fitness.

### Ethical consideration

Ethical clearance was obtained from Bahir Dar University, College of Medicine and Health Science Institutional Review Board with protocol number of 310/2021. Then, supportive letters were gained from Amhara Public Health Institute and Bahir Dar City Administration Health Office and given to concerning public health facilities before the study period to gate permission. After permission is gained from the concerning body of the institution data was extracted from the patient charts and high viral load registration book. Hence, the confidentiality of data will be kept at all levels of the study and the data will not be used for purposes other than for this study.

## Results

### Sociodemographic characteristics of study participants

The study contained 567 medical records of HIV positive, with a high viral load, and on EAC study participants and 546 (96.3%) records were included in the review. Of the study participants, majority 300(54.9%) were Females. More than two-third of the study participants 476 (87.2%) were urban residents. The mean age of study participants was 31.8(±11.54SD) years (Table 1).

### Proportion of viral load suppression

Among all HIV seropositive people who received enhanced adherence counseling 312(57.1% (95%CI; 52.9–61.3)) had achieved viral load suppression after EAC (Fig 1).

### Clinical related characteristics of the respondent

The median baseline viral load count was 8548.00 copies/ mm3 (IQR 2879.00–33645.00 copies/ml). Majority 466(85.4%) of the respondent had no recurrent opportunistic infection (Table 2).

### Treatment related characteristics of the respondents

Majority of the study participant were on first line regimen 452(82.8%) and 480(87.9%) had good ART adherence to antiretroviral therapy. The average duration of respondents on antiretroviral therapy were 112.47 months (±44.56 month). The mean time gap between high viral load detected and enhanced adherence counseling started was 56.15 days (±69.46days). The median time taken to complete EAC sessions were 141 (IQR 86–176) days (Table 3).

### Behavioral related characteristics of the participants

More than half 408(74.7%) of the study participants disclosed their HIV status. A significant number 455(83.3%) of the study participants were confident to take their antiretroviral therapy openly at home (Table 4).

**Table 1. Sociodemographic characteristics of people with high viral load and at enhanced adherence counseling at Bahir Dar city public health facility Bahir Dar, Northwest Ethiopia, 2022.** (n = 546).

| Variable | Frequency n (%) | Viral load | |
|---|---|---|---|
| | | Suppressed n (%) | Unsuppressed n (%) |
| **Age in years** | | | |
| ≤ 25 | 155(28.4) | 94(60.6) | 61(39.4) |
| 26–32 | 130(23.8) | 74(56.9) | 56(43.1) |
| 33–40 | 160(29.3) | 94(58.8) | 66(41.2) |
| ≥41 | 101(18.5) | 50(49.5) | 51(50.5) |
| **Sex** | | | |
| Male | 246(45.1) | 130(52.9) | 116(47.1) |
| Female | 300(54.9) | 182(60.7) | 118(39.3) |
| **Residency** | | | |
| Rural | 70(12.8) | 38(54.3) | 32(45.7) |
| Urban | 476(87.2) | 274(57.6) | 202(42.4) |
| **Marital status** | | | |
| Never Married | 219(40.1) | 125(57.1) | 94(42.9) |
| Married | 181(33.2) | 99(54.7) | 82(45.3) |
| Divorced | 102(18.7) | 60(58.8) | 42(41.2) |
| Widowed | 44(8.1) | 28(63.6) | 16(36.4) |
| **Religion** | | | |
| Orthodox Tewahido | 521(95.4) | 294(56.4) | 227(43.6) |
| Muslim | 16(2.9) | 11(68.7) | 5(31.3) |
| Protestant/catholic | 9(1.7) | 7(77.8) | 2(22.2) |
| **Type of Health institution** | | | |
| Health center | 220(40.3) | 123(55.9) | 97(44.1) |
| Primary hospital | 13(2.4) | 5(38.5) | 8(61.5) |
| Specialized hospital | 313(57.3) | 184(58.8) | 129(41.2) |
| **Source of referral** | | | |
| Within the health facility | 88(16.1) | 51(58.0) | 37(42.0) |
| Outside the health facility | 297(54.4) | 164(55.2) | 133(44.8) |
| Self-referral | 161(29.5) | 97(60.3) | 64(39.7) |
| **Educational Status** | | | |
| No education | 169(30.9) | 100(59.2) | 69(40.8) |
| Primary | 161(29.5) | 96(59.6) | 65(40.4) |
| Secondary and above | 216(39.6) | 116(53.7) | 100(46.3) |

## Predictors of viral load suppression

In bivariate Log-Binomial regression model variables like age, sex, religion, presence of recurrent OI, history of missing ARV doses, having non-supportive family, duration on ART, length of stay on EAC, number of ARV regimen doses taken per day, baseline viral load, number of EAC sessions, attending EAC regularly, history of interrupting ART follow-up, and sexual status were variables having a p-value of <0.25.

Variables having p-value of <0.25 on the bivariable analysis were included in the multivariable Log-Binomial regression model. Finally, it was found that religion, presence of recurrent OI, length of stay on EAC, number of ARV regimen doses taken per day, and baseline viral load were a statistically significant predictors of viral load suppression after EAC at p-value of < 0.05.

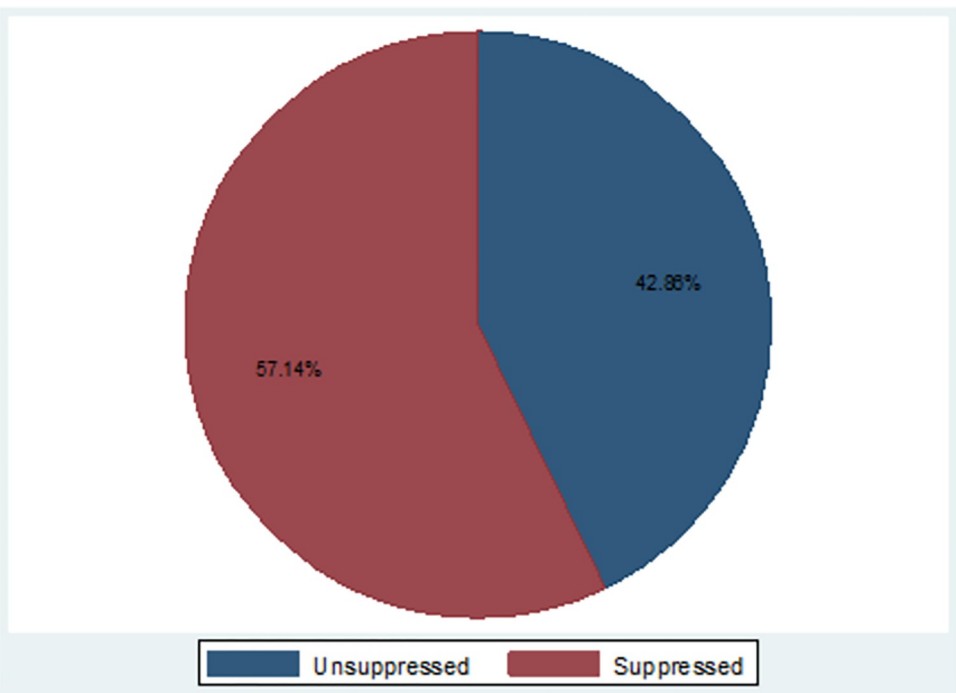

**Fig 1. Magnitude of viral load suppression of HIV seropositive people after EAC at Bahir Dar city public health facility Bahir Dar, Northwest Ethiopia, 2022.** (n = 546).

In this study, participants who had no recurrent OI were 1.40 times more likely to have viral load suppression compared to those who had recurrent OI (ARR 1.40; CI 1.03–1.91). Length of stay on EAC for less than 3 months and 3–6 months were 1.54, and 1.38 times more likely to have suppressed viral load after EAC (ARR 1.54; CI 1.19–1.99), and (ARR 1.38; CI

**Table 2. Clinical related characteristics of people with high viral load and at enhanced adherence counseling at Bahir Dar city public health facility Bahir Dar, Northwest Ethiopia, 2022.** (n = 546).

| Variable | Frequency n (%) | Viral load | |
|---|---|---|---|
| | | Suppressed n (%) | Unsuppressed n (%) |
| **Hospital admission history** | | | |
| No | 455(83.3) | 265(58.2) | 190(41.8) |
| Yes | 91(16.7) | 47(51.7) | 44(48.3) |
| **Recurrent OI** | | | |
| No | 466(85.4) | 276(59.2) | 190(40.8) |
| Yes | 80(14.6) | 36(45.0) | 44(55.0) |
| **Baseline Viral load** | | | |
| ≤2879 copies/ml | 137(25.1) | 96(70.1) | 41(29.9) |
| 2880–8548 copies/ml | 136(24.9) | 69(50.7) | 67(49.3) |
| 8549–33645 copies/ml | 137(25.1) | 76(55.5) | 61(44.5) |
| ≥33646 copies/ml | 136(24.9) | 71(52.2) | 65(47.8) |
| **Baseline BMI** | | | |
| Normal | 316(57.9) | 175(55.4) | 141(44.6) |
| Underweight | 151(27.6) | 85(56.3) | 66(43.7) |
| Overweight | 79(14.5) | 52(65.8) | 27(34.2) |

**Table 3. Treatment related characteristics of people with high viral load and at enhanced adherence counseling at Bahir Dar city public health facility Bahir Dar, Northwest Ethiopia, 2022.** (n = 546).

| Variable | Frequency n (%) | Viral load | |
|---|---|---|---|
| | | Suppressed n (%) | Unsuppressed n (%) |
| **Duration on ART in months** | | | |
| ≤ 80 | 138(25.3) | 70(50.7) | 68(49.3) |
| 81–111 | 136(24.9) | 88(64.7) | 48(35.3) |
| 112–148 | 138(25.3) | 80(58.0) | 58(42.0) |
| ≥149 | 134(24.5) | 74(55.2) | 60(44.8) |
| **ARV doses per day** | | | |
| Once | 360(65.9) | 218(60.6) | 142(39.4) |
| Twice | 186(34.1) | 94(50.5) | 92(49.5) |
| **Current ARV regimen** | | | |
| First line | 452(82.8) | 259(57.3) | 193(42.7) |
| Second or Third line | 94(17.2) | 53(56.4) | 41(43.6) |
| **ART adherence level** | | | |
| Poor | 27(5.0) | 16(59.3) | 11(40.7) |
| Fair | 39(7.1) | 25(64.1) | 14(35.9) |
| Good | 480(87.9) | 271(56.5) | 209(43.5) |
| **Missing ARV doses** | | | |
| No | 374(68.5) | 199(53.2) | 175(46.8) |
| Yes | 172(31.5) | 113(65.7) | 59(34.3) |
| **Interruption ART care follow-up** | | | |
| No | 533(97.6) | 309(58.0) | 224(42.0) |
| Yes | 13(2.4) | 3(23.1) | 10(76.9) |
| **Attend EAC regularly** | | | |
| No | 27(5.0) | 8(29.6) | 19(70.4) |
| Yes | 519(95.0) | 304(58.6) | 215(41.4) |
| **Length of stay on EAC** | | | |
| <3months | 124(22.7) | 76(61.3) | 48(38.7) |
| 3-6months | 265(48.5) | 162(61.1) | 103(38.9) |
| >6months | 157(28.8) | 74(47.1) | 83(52.9) |
| **Total number of EAC** | | | |
| ≤3 | 133(24.4) | 87(65.4) | 46(34.6) |
| 4–6 | 370(67.7) | 202(54.6) | 168(45.4) |
| ≥7 | 43(7.9) | 23(53.5) | 20(46.5) |
| **Forget to take ARV dose on time** | | | |
| No | 169(30.9) | 91(53.9) | 78(46.2) |
| Yes | 377(69.1) | 221(58.6) | 156(41.4) |

1.12–1.69) respectively. In clients who were on once per day ARV dose regimen the probability of achieving viral load suppression after EAC were 1.28 times higher as compared to those who were on twice per day ARV dose regimen (ARR 1.28; CI 1.03–1.58).

The probability of viral load suppression 1.30 times higher in participant who had baseline viral load ≤2879.00 copies/ml compared to those who have baseline viral load of ≥33646 copies/ml (ARR 1.30, CI 1.06–1.60). On the other hand, the probability of viral load suppression was 63% lower for participants who were Orthodox Tewahido Christian compared to those Protestant or Catholic Christians (ARR 0.37; CI 0.18–0.75) (Table 5).

**Table 4. Behavioral related characteristics of people with high viral load and at enhanced adherence counseling at Bahir Dar city public health facility Bahir Dar, Northwest Ethiopia, 2022.** (n = 546).

| Variable | Frequency n (%) | Viral load | |
|---|---|---|---|
| | | Suppressed n (%) | Unsuppressed n (%) |
| **HIV status disclosed** | | | |
| No | 138(25.3) | 76(55.1) | 62(44.9) |
| Yes | 408(74.7) | 236(57.8) | 172(42.2) |
| **Discontinue ARV to take other remedies** | | | |
| No | 486(89.0) | 277(57.0) | 209(43.0) |
| Yes | 60(11.0) | 35(58.3) | 25(41.7) |
| **Having non-supportive family** | | | |
| No | 479(87.7) | 267(55.7) | 212(44.3) |
| Yes | 67(12.3) | 45(67.2) | 22(32.8) |
| **Confident to take ARV openly at home** | | | |
| No | 91(16.7) | 49(53.9) | 42(46.1) |
| Yes | 455(83.3) | 263(57.8) | 192(42.2) |
| **Current sexual status** | | | |
| Not active | 280(51.3) | 171(61.1) | 109(38.9) |
| Active | 266(48.7) | 141(53.0) | 125(47.0) |
| **Utilize condom consistently** | | | |
| No | 194(72.9) | 101(52.1) | 93(47.9) |
| Yes | 72(27.1) | 40(55.6) | 32(44.4) |
| **Depressed** | | | |
| No | 437(80.0) | 251(57.4) | 186(42.6) |
| Yes | 109(20.0) | 61(56.0) | 48(44.0) |
| **Substance use** | | | |
| No | 456(83.5) | 264(57.9) | 192(42.1) |
| Yes | 90(16.5) | 48(53.3) | 42(46.7) |
| **Lack of food is a problem to take ARV** | | | |
| No | 483(88.5) | 278(57.6) | 205(42.4) |
| Yes | 63(11.5) | 34(54.0) | 29(46.0) |

## Discussion

This study was conducted to assess the outcome of the EAC on HIV seropositive people with a high viral load count. The findings of this study suggest that the overall viral load suppression after enhanced adherence counseling sessions was 57.1% (95%CI; 52.9–61.3). This finding is similar to the study done in West Gojjam Zone, Ethiopia (51.73%) [48], Nigeria (51.0%) [25], and Uganda (60%) [49]. However, this finding is lower than the findings reported by studies done in North Wollo, Ethiopia (66.4%) [14], South Africa (64%) [27], Khayelitsha, South Africa (68%) [50], Johannesburg, South Africa (64%) [27], United States of America (67.6%) [22]. Most importantly, the current finding is too far from WHO and UNAIDS's 95-95-95 ambitious plan of reaching viral load suppression to 95% by 2025 [5, 10]. This difference might be due to the difference in facility-related resources and study population. The current study includes all individuals with high viral load, whereas the previous study includes only those who were on the first-line ARV regimen. On the contrary, this finding is greater than the finding reported in the studies done in Southwestern, Ethiopia (20.3%) [51], Harare, Zimbabwe (31.2%) [26], Mumbai, India (28%) [24], Kediri City, Indonesia (9.9%) [52], a worldwide systematic review (70.5%) [53], and meta-analysis done by WHO (46.1%) [21].

**Table 5. Factors associated with viral load suppression among high viral load HIV infected people after enhanced adherence counselling session between January 2017 to December 2021 at Bahir Dar city public health facility Bahir Dar, Northwest Ethiopia, 2022.** (n = 546).

| Variables | Viral Load Status | | CRR (95% CI) | ARR (95% CI) | P-Value |
|---|---|---|---|---|---|
| | Suppressed n (%) | Unsuppressed n (%) | | | |
| **Age in Years** | | | | | |
| ≤ 25 | 94(60.6) | 61(39.4) | 1.23(0.97–1.55) | 1.16(0.91–1.49) | 0.233 |
| 26–32 | 74(56.9) | 56(43.1) | 1.15(0.90–1.47) | 1.03(0.79–1.35) | 0.829 |
| 33–40 | 94(58.8) | 66(41.2) | 1.19(0.94–1.50) | 1.12(0.90–1.38) | 0.301 |
| ≥41 | 50(49.5) | 51(50.5) | 1 | 1 | |
| **Sex** | | | | | |
| Male | 182(60.7) | 118(39.3) | 1.15(0.99–1.33) | 1.08(0.94–1.25) | 0.288 |
| Female | 130(52.9) | 116(47.1) | 1 | 1 | |
| **Recurrent OI** | | | | | |
| No | 276(59.2) | 190(40.8) | 1.32(1.02–1.70) | 1.40(1.03–1.91) | **0.033** |
| Yes | 36(45.0) | 44(55.0) | 1 | 1 | |
| **Having non-supportive family** | | | | | |
| No | 267(55.7) | 212(44.3) | 0.83(0.69–0.99) | 0.96(0.74–1.24) | 0.732 |
| Yes | 45(67.2) | 22(32.8) | 1 | 1 | |
| **Duration on ART** | | | | | |
| ≤ 80 months | 70(50.7) | 68(49.3) | 0.92(0.73–1.15) | 0.85(0.67–1.07) | 0.161 |
| 81-111months | 88(64.7) | 48(35.3) | 1.17(0.96–1.43) | 1.08(0.89–1.31) | 0.430 |
| 112–148 months | 80(58.0) | 58(42.0) | 1.05(0.85–1.29) | 0.87(0.69–1.10) | 0.222 |
| ≥149 months | 74(55.2) | 60(44.8) | 1 | 1 | |
| **Length of stay on EAC** | | | | | |
| <3months | 76(61.3) | 48(38.7) | 1.30(1.05–1.62) | 1.54(1.19–1.99) | **0.001** |
| 3-6months | 162(61.1) | 103(38.9) | 1.30(1.07–1.57) | 1.38(1.12–1.69) | **0.002** |
| >6months | 74(47.1) | 83(52.9) | 1 | 1 | |
| **ARV dose per day** | | | | | |
| Once | 218(60.6) | 142(39.4) | 1.20(1.02–1.41) | 1.28(1.03–1.58) | **0.023** |
| Twice | 94(50.5) | 92(49.5) | 1 | 1 | |
| **Baseline viral load** | | | | | |
| ≤2879 copies/ml | 96(70.1) | 41(29.9) | 1.34(1.10–1.63) | 1.30(1.06–1.60) | **0.012** |
| 2880–8548 copies/ml | 69(50.7) | 67(49.3) | 0.97(0.77–1.22) | 0.99(0.77–1.26) | 0.909 |
| 8549 - 33645copies/ml | 76(55.5) | 61(44.5) | 1.06(0.85–1.32) | 1.08(0.85–1.36) | 0.535 |
| ≥33646copies/ml | 71(52.2) | 65(47.8) | 1 | 1 | |
| **Religion** | | | | | |
| Orthodox Tewahido | 294(56.4) | 227(43.6) | 0.73(0.51–1.04) | 0.37(0.18–0.75) | **0.006** |
| Muslim | 11(68.7) | 5(31.3) | 0.88(0.55–1.43) | 0.52(0.24–1.16) | 0.109 |
| Protestant/Catholic | 7(77.8) | 2(22.2) | 1 | 1 | |
| **Current sexual status** | | | | | |
| Not active | 171(61.1) | 109(38.9) | 1.15(0.99–1.33) | 1.05(0.89–1.24) | 0.541 |
| Active | 141(53.0) | 125(47.0) | 1 | 1 | |

This study revealed that religion, presence of recurrent OI, length of stay on EAC, number of ARV regimen doses taken per day, and baseline viral load were statistically significant predictors of viral load suppression after EAC.

This study finding indicated that the probability of viral load suppression is higher in those who had no recurrent opportunistic infection as compared to those who had. This is could due

to the fact that, the absence of recurrent opportunistic infection indicates that the client has somehow intact immunity and this will help the immune system to suppress viral replication [10, 12]. Additionally, clients with recurrent opportunistic infection will be exposed to different drugs which will further increase the tablet burden and lead to poor adherence. This finding is consistent with the studies done in Ethiopia [37, 48, 54], South Africa [55], Uganda [56], and USA [22].

The current study showed that the shorter length of stay on EAC increase the probability of viral load suppression after EAC. This could be explained by, clients who complete their EAC within a given standard time frame (within six months) most likely will have good adherence this intern lead to viral load suppression. Therefore, having a short length of stay indicates that the probability of having suppressed viral load is high [10, 12]. It is well known that the recommended number of EAC sessions is 3–6 which will be conducted every month. However, WHO recommends that if the patient is still not adhering to his treatment after 6 months of EAC, the EAC sessions could be continued until they adhere. This result is in line with another study done in Zimbabwe [26].

This study finding indicated that the probability of viral load suppression is increased in those who have the lowest baseline viral load as compared to those who had the highest baseline viral load. This could be explained by the fact that clients having the lowest baseline viral load count ≤2879 copies/ml need less time to achieve suppressed viral load as compared to their counterparts (viral load ≥33646copies/ml).This finding is consistent with the studies done in North Wollo, Ethiopia [14], Kombolcha Town, Ethiopia [37], Harare, Zimbabwe [26], and Rakai, Uganda [49].

The current study also found that being Orthodox Tewahido Christian decreases the probability of achieving viral load suppression after EAC. It might be justified that individual might discontinue their follow-up to take other remedies like holy water. This is consistent with the participant's report, in which their main reason for interrupting follow-up and missing ARV dose was their intention to go to holy water treatment. However, this finding is not in line with other studies that report religion is not a predictor for viral load suppression [14, 32, 49]. On the contrary the study done in southeastern USA reported that church attendance is protective for unsuppressed viral load count [57]. Similarly, studies showed that while higher spirituality has been found to be associated with returning to HIV care in US settings [58]. This difference could be due to that the majority of the study participants in the current study were Orthodox Tewahido Christians and they believe Holy water gives cure for HIV. Consequently, they discontinue their ARV medication and follow up for Holy water service. This is supported by a study finding which reported that in some non-Western regions, spirituality has shown associations with concurrent use of alternative therapies and less adherence to antiretroviral treatments [59].

Importantly, this study also reported that those who were taking their ARV dose regimen based on a once-per-day schedule were more likely to have suppressed viral load as compared to those who were on twice per day schedule. This could be explained by the participants who were on twice per day ARV dose schedule increase the ARV tablet burden and this may lead to interruption of taking ARV doses as per the schedule. This finding is supported by the studies done in Houston, Texas [60], New Jersey Medical School in New York [61], and South Carolina [62] (USA), south Korea [63]. These finding suggest that once ARV dose regimen could improve medication adherence and the viral load suppression by decreasing pills burden.

On the contrary in the current study ARV regimen, sex, and adherence level were found not a significant predictors of viral load suppression after enhanced adherence counseling. This might indicate the need for use of better assessment methods of adherence to drugs rather than the pharmacy refill methods.

Finally, this study reported that the majority of the clients achieved viral load suppression after EAC. However, this finding is still far from the WHO and UNAIDS's target plan on viral load suppression [1]. Most importantly, the current findings support WHO recommendations that suspected virological failure should be addressed by enhanced adherence counseling as well as repeat measurement before consideration of a treatment switch to a second-line or third-line regimen [1, 10]. Thus, enhanced adherence counseling interventions can preserve the first-line treatment regimen. This could decrease health care costs and the transmission of resistant strains from newly infected people.

## Strength and limitation of the study

### Strength of the study

The study was conducted in all public health institution that provides ART service which increases the external validity. Furthermore, this study includes all high viral load patients for the last 5 years. Even though, the study used secondary data the data were collected from the patient chart, follow-up chart, and high viral load registration books, which are the primary level of documentation of the patient's information in the country.

### Limitation of the study

The study used secondary data extracted from the client's medical charts and high viral load registration book. Therefore, analysis was limited to only those variables that are recorded in the patient charts, and the high viral load registration book. Potential predictor variables like spouse education status, wealth index, smoking status, and lived experience of the clients were missed.

## Conclusion and recommendation

### Conclusion

This study showed that more than half of the study participant achieved viral load suppression after EAC. However, this finding is still far from the UNAIDS target. Religion, absence of recurrent OI, short time taken to complete EAC sessions, ARV dose regimen taken once per day, and baseline viral load $\leq 2879$ copies/ml were statistically significant predictors of viral load suppression after EAC.

### Recommendations

The Amhara Regional Health Bureau has implemented the Enhanced Adherence Counseling program to achieve the UNAIDS viral load suppression plan among all people living with HIV to 95%. However, the initiative is far from accomplishing its aim and needs to be improved in public health facilities in Bahir Dar City. The bureau should also explore techniques to lessen tablet burden and boost viral load suppression, as well as engage Orthodox Tewahido religious leaders in ART treatment initiatives. Future research should incorporate prospective follow-up studies to analyze characteristics relating to health staff, facilities, and the client, including behavioral aspects and religious beliefs regarding ARV.

## Supporting information

**S1 Appendix. Information sheet.**
(PDF)

**S2 Appendix. Data collection tools.**
(PDF)

**S3 Appendix. Declaration form.**
(PDF)

**S4 Appendix. Examiner's approval form.**
(PDF)

## Acknowledgments

First and foremost, I want to thank the faculties at Bahir Dar University, the College of Medicine and Health Sciences, the School of Public Health, and the School of Health Sciences for their unwavering support in helping me complete this research project. Second, I'd like to express my gratitude for the assistance provided during the data collection period by the employees of the Bahir Dar municipal health department, the ART focal points, and the data clerks who have been employed in each of the chosen public health institutions. Without a doubt, Mr. Zelalem Mehari Nigussie deserves special recognition for his unwavering assistance with the data analysis.

## Author Contributions

**Conceptualization:** Minyichil Birhanu Belete, Abebayehu Bitew, Kebadnew Mulatu.

**Data curation:** Minyichil Birhanu Belete.

**Formal analysis:** Minyichil Birhanu Belete.

**Investigation:** Minyichil Birhanu Belete.

**Methodology:** Minyichil Birhanu Belete, Abebayehu Bitew, Kebadnew Mulatu.

**Supervision:** Minyichil Birhanu Belete, Abebayehu Bitew, Kebadnew Mulatu.

**Validation:** Abebayehu Bitew, Kebadnew Mulatu.

**Writing – original draft:** Minyichil Birhanu Belete.

**Writing – review & editing:** Minyichil Birhanu Belete, Abebayehu Bitew, Kebadnew Mulatu.

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
