## [Decision Letter · Decision Letter 0]

20 Nov 2023

PONE-D-23-34729VIRAL LOAD SUPPRESSION AND ITS PREDICTOR AMONG HIV SEROPOSITIVE PEOPLE WHO RECEIVE ENHANCED ADHERENCE COUNSELING AT PUBLIC HEALTH INSTITUTIONS IN BAHIR DAR, NORTHWEST ETHIOPIA RETROSPECTIVE FOLLOW-UP STUDYPLOS ONE

Dear Dr. Belete,

Thank you for submitting your manuscript to PLOS ONE. After careful consideration, we feel that it has merit but does not fully meet PLOS ONE’s publication criteria as it currently stands. Therefore, we invite you to submit a revised version of the manuscript that addresses the points raised during the review process.

We look forward to receiving your revised manuscript.

Kind regards,

Chalie Tiruneh Marew, MSc

Academic Editor

PLOS ONE

Journal Requirements:

**Additional Editor Comments:**

Intensive English Language editing is required.

Follow the standards of the journal in writing the abstract.

The background section is too bulky; make it to the point and summarize in not more than 2-3 pages.

Include the documents listed in the annex section of your supplementary file.

Generally, the manuscript is bulky; it needs to be presented in a short and concise way

Reviewers' comments:

Reviewer's Responses to Questions

**Comments to the Author**

1. Is the manuscript technically sound, and do the data support the conclusions?

Reviewer #1: Yes

Reviewer #2: Yes

2. Has the statistical analysis been performed appropriately and rigorously? 

Reviewer #1: Yes

Reviewer #2: Yes

3. Have the authors made all data underlying the findings in their manuscript fully available?

Reviewer #1: Yes

Reviewer #2: No

4. Is the manuscript presented in an intelligible fashion and written in standard English?

Reviewer #1: Yes

Reviewer #2: Yes

5. Review Comments to the Author

Reviewer #1: Comments

Thank you for the opportunity to review the manuscript entitled “viral load suppression and its predictors among HIV-seropositive who received Enhanced adherence counseling at public health institution in Bahir Dar, Northwest Ethiopia. Retrospective follow-up study.

I think this study provides information predictors among HIV-seropositive who received Enhanced adherence counseling at public health institution, and this study might be an excellent input for the national government for decision making.

Please see the comments and respond accordingly below before considering this manuscript suitable for publication.

I do have an issue on your title; many studies have carried out related to these topics what makes add for the scientific communities.

From the abstract section the background section is too long you should have state your research question or your problem 2 to 3 statements

In your conclusion part you have described that “the majority of individuals had decreased viral loads” what is your parameters to say decreased or have done any comparison group

Generally your abstract section too long, make it short as much as you possible.

From your introduction section you have wrote greater than seven pages, since it’s a scientific paper you should have narrate to one page unless maximum two page by focusing your research question and significance your study, unless its seems a negligence of the authorship.

On your document on page 4, 6 &24 your stating “In September 2020, UNAIDS reported a gap in achieving its 90-90-90 global targets”, its an old evidence wrote here please the current evidence on UNAIDS global target plan.

I doubt a clarity question about on your methodology section especially your study design (really retrospective follow-up study or not?? ), if its right I have a lot question rises on your variables, where you do get behavioral related question and others?

Your study, target population is does not clearly explained (adult, adolescence, or pediatrics age group)

Could you think its enough single population proportion formula for sample size determination, if not why you are not stated here?

I think it’s better to write or define enhanced adherence counseling(EAC) on your operational definition

Again Behavioral related predictors(Cigarette smoking, alcohol drinking, khat chewing) were present in the intake form as well in the follow-up form, even also its difficult addressed from your Treatment related factors such as time gap from high viral load detected to EAC session start, time to complete EAC sessions, history of drug discontinuation. Its A big Issue for me needs intense response unless it might be return back re-analysis.

If your data collection were in electronic based platform what its need of supervisor, how do you managed information contamination since you have shared your data for your data collectors and supervisors?

Since you have generated or collect five years retrospective data how much really your variables is feasible to avail such variables Source of referral variables??Depressed???

Am happy the way you state result section

Minor comment on page 17 sub topics “Proportion of unsuppressed Viral Load” makes it suppressed unless the body of the statement seems paradox, or make it correct the statement.

One of your predictors for viral load suppression is “being Orthodox Tewahido Christian” which is “decreases the probability of achieving viral load suppression after EAC”, so what do you recommend to scientific community, international community. In my point of view this variables is not necessary to state here as predictors. Unless its strong evidence to recommendation.

Your conclusion “ this study reported that viral load suppression after EAC was found high”, what is your evidence to say “HIGH”

Reviewer #2: Title: your title is not novel, because there are recent similar studies conducted in north Wollo and west Gojam.

Abstract: the number of words in this section is beyond the scientifically acceptable limit and it should be summarized in one page.

Introduction: it is vague (more than 7 pages) and seeming literature review, please try to summarize with no more than 4 pages in concise manner.

Methods and materials: in your study area part better to describe how many HIV sero-positive people are served and reduce other unnecessary statements especially those stated in line no 220-222. Are your study, source and target population similar? If yes, write them separately.

Result and Discussion: Good, but I doubt whether all the significant variables of the study are discussed properly since they are excess, i.e., six.

In the result section make some of the tables in figure form to make the study more attractive.

Strength and Limitation: the same idea is written on line number 454 and 458. Better to consider it as limitation of your study.

Conclusion and recommendation: conclude your outcome variable, like; suppressed or not suppressed, low or high. The recommendation should be summarized with no more than three sentences.

General comments:

- Use the scientifically recommended font size all over your document.

- Take grammatical corrections for some sections of your document.

- Totally the manuscript is vague, so please try to make it concise based on the given document; especially the introduction section.

6. PLOS authors have the option to publish the peer review history of their article (what does this mean?). If published, this will include your full peer review and any attached files.

Reviewer #1: No

Reviewer #2: **Yes: **Adane Birhanu Nigat

---

## [Author Response · Author response to Decision Letter 0]

25 Jan 2024

Response to Reviewers

Reviewer #1

I do have an issue on your title; many studies have carried out related to these topics what makes add for the scientific communities.

Response:

There are a few studies in the area of viral load suppression after EAC. However, there is not a single study in our study area. Additionally, some of the papers are focused on the adult age group only. Our study area was a specialized referral hospital that could include a population on the 3rd ART regimen, whereas the previous studies focused on HIV-infected people on the first-line regimen only.

Reviewer #1

From the abstract section the background section is too long you should have state your research question or your problem 2 to 3 statements

Response: 

We try to summarize the background session of the abstract 

Reviewer #1

In your conclusion part you have described that “the majority of individuals had decreased viral loads” what is your parameters to say decreased or have done any comparison group

Response: 

We say the majority based on the finding that we go which was 312 (57.1%) which is more than the half of the participants 

Reviewer #1

From your introduction section you have wrote greater than seven pages, since it’s a scientific paper you should have narrate to one page unless maximum two page by focusing your research question and significance your study, unless it seems a negligence of the authorship.

Response

Thank you for your feedback on my paper. I appreciate your suggestion regarding the length of the introduction section. I understand that scientific papers typically have shorter introduction sections and I apologize for any inconvenience my longer introduction may have caused you.

I assure you that I have not neglected my responsibilities as an author and have taken great care to ensure that my paper is of high quality and meets the standards of the journal and I resynthesis the introduction part. I appreciate your time and effort in reviewing my paper and welcome any further feedback or suggestions you may have.

Reviewer #1

I doubt a clarity question about on your methodology section especially your study design (really retrospective follow-up study or not??), if its right I have a lot question rises on your variables, where you do get behavioral related question and others?

Response: 

My study design was a retrospective follow-up study, and the standardized adherence counseling format incorporates variables regarding behavioral-related questions such HIV status disclosure status, quitting ARV to take alternative treatments, Having a non-supportive family, being confident to take ARV openly at home, having a current sexual status, condom utilization status, depression status, substance use, and a lack of food are obstacles with taking ARV, etc. associated difficulties are clearly indicated in the adherence counseling, which will be filled out in all appointments.

Reviewer #1

Your study, target population is does not clearly explained (adult, adolescence, or pediatrics age group)

Response 

The target population of our study was all age group HIV seropositive people received enhanced adherence counseling at public health facilities

Reviewer #1

Could you think it’s enough single population proportion formula for sample size determination, if not why you are not stated here?

Response

At the design stage of our study, we compute our sample size using both the double population proportion formula and the single population proportion formula, but we get the greatest sample size when we utilize the single population proportion calculation. And that is why we put only the formula that we last utilized.

Reviewer #1

I think it’s better to write or define enhanced adherence counseling (EAC) on your operational definition

Response 

We operationalize what is Enhanced Adherence Counseling 

Reviewer #1

Again, Behavioral related predictors (Cigarette smoking, alcohol drinking, khat chewing) were present in the intake form as well in the follow-up form, even also its difficult addressed from your Treatment related factors such as time gap from high viral load detected to EAC session start, time to complete EAC sessions, history of drug discontinuation. Its A big Issue for me needs intense response unless it might be return back re-analysis.

Response 

We collect our data from the high viral load registration booklet, intake form, EAC follow-up form, and client medical card. With this all-data source, we assure that we can access all the data related to behavioral-related predictors (cigarette smoking, alcohol drinking, and khat chewing), treatment-related factors such as the time gap from high viral load detected to EAC session start, time to complete EAC sessions, and history of drug discontinuation. All this data is readily available from the above-mentioned data sources that we employ.

Reviewer #1

If your data collection were in electronic based platform what its need of supervisor, how do you managed information contamination since you have shared your data for your data collectors and supervisors?

Response 

We collect data using an electronic platform, and supervisors were recruited to observe the quality of data at each data collection site and to examine the daily data collected by each data collector before the data is uploaded to the server.

Regarding the information contamination, information contamination is not a concern for us since we are using secondary data. 

Reviewer #1

Since you have generated or collect five years retrospective data how much really your variables is feasible to avail such variables Source of referral variables??Depressed??? 

Response 

As the authors of this study, we ensure that all the data characteristics were assured as much as feasible. Additionally, Enhanced Adherence Counseling for HIV seropositive patients with a high viral load is a well-structured program that is actively supported, and data quality is monitored and assessed by CDC professionals recruited there. With all these guidelines, we can assure that all the data we utilize is clearly documented in the data source that we use.

Reviewer #1

One of your predictors for viral load suppression is “being Orthodox Tewahido Christian” which is “decreases the probability of achieving viral load suppression after EAC”, so what do you recommend to scientific community, international community. In my point of view these variables is not necessary to state here as predictors. Unless its strong evidence to recommendation.

Response 

Based on our findings, the majority of the study participants were Orthodox Tewahido Christians, and they believe Holy Water gives a cure for HIV. Consequently, they discontinue their ARV medication and follow up for Holy Water service. As evidenced by the EAC follow-up form, the majority of those clients who discontinued ART were to go to Holy Water Service. So, our recommendation is to counsel them so they can take ART in parallel with holy water instead of discontinuing the drug. And we put our recommendation in the recommendation session on lines 488 to 489 as “And also, it is better to strengthen the engagement of Orthodox Tewahido religious leaders in ART care programs.”

Reviewer #1

Your conclusion “this study reported that viral load suppression after EAC was found high”, what is your evidence to say “HIGH”

Response 

We see your concern, and it is a typographic error. We consider your feedback and correct it accordingly. 

Reviewer #2: 

Title: your title is not novel, because there are recent similar studies conducted in north Wollo and west Gojam.

Response 

We cited both research findings in our study; however, the study in the north Wollo zone focused primarily on HIV-infected patients on the first-line regimen. Whereas the study done in west Gojam focused mainly on HIV-positive adult patients on ART. Thus, the current study covers clients in all regimens and all age categories. 

Reviewer #2: 

Abstract: the number of words in this section is beyond the scientifically acceptable limit and it should be summarized in one page.

Response 

We welcome your suggestion, and we summarize the abstract part accordingly.

Reviewer #2: 

Introduction: it is vague (more than 7 pages) and seeming literature review, please try to summarize with no more than 4 pages in concise manner.

Response

We accept your valuable feedback and we amend accordingly 

Reviewer #2: 

Methods and materials: in your study area part better to describe how many HIV sero-positive people are served and reduce other unnecessary statements especially those stated in line no 220-222. Are your study, source and target population similar? If yes, write them separately.

Response 

In our research, the target population and study population were not similar, and we described them separately according to your recommendation.

Reviewer #2: 

Result and Discussion: Good, but I doubt whether all the significant variables of the study are discussed properly since they are excess, i.e., six.

Response

Thank you for your feedback on the results and discussion section of our study. We appreciate your comment on the number of significant variables, and we would like to clarify that all variables were discussed in detail, although we understand that there may be an excess.

In our study, we aimed to investigate the relationship between several variables and the outcome of interest. Through our analysis, we found five variables to be significant predictors of the outcome. In our discussion, we provided a comprehensive overview of each significant variable, including its theoretical relevance, effect size, and direction of association with the outcome. We also discussed the potential mechanisms underlying these associations and their implications for future research and practice.

However, we understand that the discussion of multiple variables can be complex and overwhelming. To address this issue, we have revised our discussion section to provide a more concise and focused overview of each variable, highlighting their unique contributions to the outcome. We hope that this revised version better addresses your concerns and provides a clearer understanding of our findings. Thank you again for your valuable feedback. 

Reviewer #2: 

In the result section make some of the tables in figure form to make the study more attractive.

Strength and Limitation: the same idea is written on line number 454 and 458. Better to consider it as limitation of your study.

Response 

When evaluating our research, we identified the data sources as a key strength and highlighted them as the primary level of documentation. However, despite their importance, we acknowledged that they are not without flaws, which we discussed in the limitations section. It's worth noting that the concepts we covered in the strength and limitation parts are distinct from one another.

Reviewer #2: 

Conclusion and recommendation: conclude your outcome variable, like; suppressed or not suppressed, low or high. The recommendation should be summarized with no more than three sentences.

Response 

We accept your feedback, and we amend accordingly.

---

## [Decision Letter · Decision Letter 1]

20 Feb 2024

PONE-D-23-34729R1Viral load suppression and its predictor among HIV seropositive people who receive enhanced adherence counseling at public health institutions in Bahir Dar, Northwest Ethiopia. Retrospective follow-up study

PLOS ONE

Dear Dr. Belete,

Thank you for submitting your manuscript to PLOS ONE. After careful consideration, we feel that it has merit but does not fully meet PLOS ONE’s publication criteria as it currently stands. Therefore, we invite you to submit a revised version of the manuscript that addresses the points raised during the review process.

A rebuttal letter that responds to each point raised by the academic editor and reviewer(s). You should upload this letter as a separate file labeled 'Response to Reviewers'.A marked-up copy of your manuscript that highlights changes made to the original version. You should upload this as a separate file labeled 'Revised Manuscript with Track Changes'.An unmarked version of your revised paper without tracked changes. You should upload this as a separate file labeled 'Manuscript'.We look forward to receiving your revised manuscript.

Kind regards,

Chalie Marew Tiruneh, MSc

Academic Editor

PLOS ONE

Journal Requirements:

**Additional Editor Comments:**

**ACADEMIC EDITOR:**

Dear Authors!

Based on the feedback provided by the reviewers and editor, it is necessary to revise the background of the study before we can move forward to the next step. The revised background should provide a clear and concise summary of the main background, considering the comments and suggestions provided by the reviewers and editor.

Reviewers' comments:

Reviewer's Responses to Questions

**Comments to the Author**

Reviewer #1: 

2. Is the manuscript technically sound, and do the data support the conclusions?

Reviewer #1: Yes

3. Has the statistical analysis been performed appropriately and rigorously? 

Reviewer #1: Yes

4. Have the authors made all data underlying the findings in their manuscript fully available?

Reviewer #1: Yes

5. Is the manuscript presented in an intelligible fashion and written in standard English?

Reviewer #1: Yes

6. Review Comments to the Author

Reviewer #1: Most of the previous comments are well addressed, but from the introduction section is not answered well, unless revised or amend it I regret to procced to next step (accept for publication). So please narrate the introduction section atmost to 2 to 4 pages.

7. PLOS authors have the option to publish the peer review history of their article (what does this mean?). If published, this will include your full peer review and any attached files.

**Do you want your identity to be public for this peer review?** 

Reviewer #1: No

---

## [Author Response · Author response to Decision Letter 1]

28 Feb 2024

Response to Reviewers

Journal editor:

Response: 

We made amendments on some reference list 

Rfererence number 4

“4. World Health Organization. Consolidated guidelines on HIV prevention, diagnosis, treatment and care for key populations: 2016 update. 2016.” in the previous version is amended as:

“7. World Health Organization. Consolidated guidelines on HIV prevention, diagnosis, treatment and care for key populations: World Health Organization; 2016. Available from: https://www.who.int/publications/i/item/9789241511124.”

Rfererence number 9

“9. World Health Organization. Guidelines: updated recommendations on HIV prevention, infant diagnosis, antiretroviral initiation and monitoring. World Health Organization, 2021 9240022236.” in the previous version is amended as:

“10. World Health Organization. Updated recommendations on HIV prevention, infant diagnosis, antiretroviral initiation and monitoring: World Health Organization; 2021. Available from: https://www.who.int/publications/i/item/9789240022232.”

Rfererence number 11

“11. Federal Ministry of Health Ethiopia. National Consolidated Guidelines for Comprehensive Hiv Prevention, Care and Treatment. Ethiopia, Addis Ababa: Federal Ministry of Health, Ethiopia; 2018.” in the previous version is amended as:

“12. Federal Ministry of Health Ethiopia. National Consolidated Guidelines for Comprehensive Hiv Prevention, Care and Treatment. Ethiopia, Addis Ababa: Federal Ministry of Health, Ethiopia; 2018. Available from: https://www.afro.who.int/publications/national-consolidated-guidelines-comprehensive-hiv-prevention-care-and-treatment.”

Rfererence number 12

“12. World Health Organization. Consolidated guidelines on the use of antiretroviral drugs for treating and preventing HIV infection: recommendations for a public health approach. World Health Organization,, 2016 9241549688.” in the previous version is amended as:

“4. World Health Organization. Consolidated guidelines on the use of antiretroviral drugs for treating and preventing HIV infection: recommendations for a public health approach: World Health Organization; 2016. Available from: https://www.who.int/publications/i/item/9789241549684.”

Reviewer #1:

Most of the previous comments are well addressed, but from the introduction section is not answered well, unless revised or amend it I regret to procced to next step (accept for publication). So please narrate the introduction section at most to 2 to 4 pages.

Response:

Thank you for your feedback on my paper. We appreciate your suggestion regarding the length of the introduction section. 

Considering the feedbacks given by you we try to summaries and resynthesis the introduction part within the given page limit. We appreciate your time and effort in reviewing my paper and welcome any further feedback or suggestions you may have.

---

## [Editor Report · Decision Letter 2]

23 Apr 2024

Viral load suppression and its predictor among HIV seropositive people who receive enhanced adherence counseling at public health institutions in Bahir Dar, Northwest Ethiopia. Retrospective follow-up study

PONE-D-23-34729R2

Dear Dr. Belete,

We’re pleased to inform you that your manuscript has been judged scientifically suitable for publication and will be formally accepted for publication once it meets all outstanding technical requirements.

Kind regards,

Chalie Marew Tiruneh 

Academic Editor

PLOS ONE

---

## [Editor Report · Acceptance letter]

26 Apr 2024

PONE-D-23-34729R2 

PLOS ONE

Dear Dr. Belete, 

I'm pleased to inform you that your manuscript has been deemed suitable for publication in PLOS ONE. Congratulations! Your manuscript is now being handed over to our production team.

Kind regards, 

on behalf of

Mr Chalie Marew Tiruneh 

Academic Editor

PLOS ONE